# Hydrodynamic Characteristics Study of Bionic Dolphin Tail Fin Based on Bidirectional Fluid–Structure Interaction Simulation

**DOI:** 10.3390/biomimetics10010059

**Published:** 2025-01-16

**Authors:** Ning Wang, Yu Zhang, Linghui Peng, Wenchuan Zhao

**Affiliations:** College of Mechanical Engineering, Shenyang University of Technology, Shenyang 110870, China; zhangyu@sut.edu.cn (Y.Z.); lhpeng@smail.sut.edu.cn (L.P.); zhao_wenchuan@126.com (W.Z.)

**Keywords:** asymmetric motion, bionic tail fin, bidirectional fluid–structure interaction, flexibility, hydrodynamic characteristics

## Abstract

Using bidirectional fluid–structure interaction technology, the dorsal–ventral motion of the dolphin tail fin was simulated, and the feasibility of the numerical simulation method was validated through underwater motion experiments. This study investigated the effects of structural parameters and motion modes of bionic dolphin tail fins on their propulsion performance. The results show that flexible tail fins can enhance propulsion performance. Compared to equal-thickness flexible tail fins, variable-thickness flexible tail fins that conform to the structural characteristics of real dolphin tail fins exhibit better propulsion performance. Asymmetric motion modes have a certain thrust-enhancing effect, but altering the frequency ratio F and amplitude ratio H of heaving motion leads to an increase in pitching moment, reducing swimming stability. Additionally, the greater the difference in frequency and amplitude between the up-and-down motions, the larger the pitching moment. The study results provide references for the optimized design and motion control of bionic tail fins.

## 1. Introduction

With the gradual depletion of terrestrial resources, the pace of human development of marine resources has accelerated in response to increasing demand. Underwater robots, as important tools for human exploration of the ocean, have broad application prospects. Traditional propeller-driven underwater robots have disadvantages such as high noise, poor stability, and low propulsion efficiency [1,2]. Aquatic organisms in the ocean have developed unique swimming methods and extraordinary swimming abilities through long-term natural evolution. Researchers are studying the propulsion mechanisms of various underwater organisms, hoping to develop bionic underwater robots that are highly efficient, low-noise, and highly maneuverable. According to differences in morphological characteristics and swimming methods, fish movement in the ocean can be divided into body/caudal fin (BCF) propulsion mode and median/paired fin (MPF) propulsion mode [3,4]. Approximately 80% of fish in the ocean employ the BCF propulsion mode, which is characterized by fast swimming speed and high propulsion efficiency, and it has become an important subject for researchers conducting bionic engineering studies [5].

Over the past few decades, researchers both domestically and internationally have conducted extensive studies on the phenomena and mechanisms of fish swimming. Taylor [6] established the “drag model” for studying fish swimming, using steady-state theory to calculate the hydrodynamic forces experienced by the fish at a specific moment during swimming. Due to the neglect of inertial forces, this method is only applicable to low-Reynolds-number conditions. Lighthill [7,8,9,10] was the first to propose the elongated-body theory (EBT) for investigating the propulsion mechanisms of fish swimming. This theory posits that the total energy required for swimming originates from the active movement of the fish body and does not account for the influence of wake vortices on swimming performance. Wu Yaozu [11] proposed the two-dimensional waving plate theory (2DWPT) to describe the propulsion mechanisms of fish, utilizing two-dimensional potential flow theory. This theory allows for the calculation of thrust generated by swimming, the power required, the energy transferred to the wake, and propulsion efficiency. Cheng Jianyu, Tong Binggang, and others [12,13] extended the two-dimensional waving plate model to three dimensions, establishing the three-dimensional waving plate theory (3DWPT) using linear unsteady potential flow theory. This model not only takes into account the effects of inertial forces and leading-edge suction but also considers the influence of wake vortices on the waving plate. However, it is only applicable to inviscid potential flows and small-amplitude oscillations. In addition to theoretical analysis, researchers have conducted numerous experiments to study fish swimming. Gray [14] proposed the famous “Gray Paradox” through experiments and observations, which states that if the resistance experienced by a swimming dolphin were equal to that of a rigid dolphin model being towed, then the dolphin’s muscles would need to supply at least seven times the power that normal mammalian muscles can provide. This is obviously unreasonable; however, studies have found that marine organisms employ various drag reduction mechanisms during swimming, which may provide an explanation for the “Gray Paradox”. With the development of digital particle image velocimetry (DPIV) [15,16], researchers have obtained more direct means to measure and analyze the flow fields during fish swimming. Fish [17] directly measured the flow field generated during dolphin swimming using DPIV technology. Nauen and Lauder [18] studied the wake flow field of mackerel and found that the wake consists of a series of interconnected elliptical vortex rings, each containing a central jet. Muller et al. [19] employed DPIV technology to conduct detailed studies and analyses of the flow field vortex structures during the C-shaped rapid start, periodic forward swimming, and the transition from periodic forward swimming to gliding in zebrafish larvae under the sub-carangid mode.

Theoretical analysis often requires simplifying the swimming process, while experimental studies face issues such as uncontrollable experimental subjects or constraints imposed by experimental technical conditions. With the development of computational fluid dynamics (CFD) [20,21,22], numerical simulation techniques have, to some extent, compensated for the limitations of theoretical analysis and experimental research. Wolfgang et al. [23] employed the boundary element method (BEM) to conduct numerical studies on the straight-line swimming and C-shaped turning processes of a large tuna numerical model, obtaining two-dimensional velocity vector distribution maps and pressure distribution contour maps of the flow field. Zhu et al. [24], based on the BEM method, investigated the interference between vortices generated by the body and those generated by the tail fin in the swimming process of tuna adopting the BCF propulsion mode. The results indicated that the interference between body-generated vortices and tail-fin-generated vortices can enhance the propulsion efficiency of the tail fin. Li et al. [25] used a moving mesh approach to numerically study the hydrodynamic performance of pufferfish during straight-line swimming under the combined propulsion modes of boxfish and scorpionfish. The results demonstrated that bionic fish propelled by flexible fins achieve cruising speeds 1.6 to 2.0 times faster than those of fish propelled by rigid fins. Zhou et al. [26] investigated the effects of spanwise flexibility on the swimming performance of tail fins through numerical simulations. Their results indicated that spanwise flexibility can enhance thrust and improve propulsion efficiency. Ramesh et al. [27], using Fluent’s fluid–structure interaction techniques, studied the impact of phase angle and flexible tail fin design on propulsion efficiency. Their findings demonstrated that a phase angle of 90° yields the optimal propulsion performance, while flexible designs significantly enhance propulsion efficiency and reduce fluid resistance. Marvin Wright [28] analyzed the thrust performance of fixed and pitching elastic plates across a parameter space considering different material stiffnesses and pitching frequencies. The results provide valuable insights into the transient hydrodynamics and thrust generation of flexible appendages (such as BCF propulsion) and highlight performance differences under various material properties and driving parameters. Mannam et al. [29] examined the thrust generation and efficiency of flapping wings under different parameter variations through experiments and numerical simulations, offering a comparative analysis of the hydrodynamic performance between rigid and flexible flapping wings. Su et al. [30] investigated the hydrodynamic performance of a tuna-inspired robotic swimmer and conducted a detailed analysis of how flexible tail fin motion parameters influence the robot’s swimming performance.

In summary, researchers have conducted extensive studies on fish swimming through theoretical analysis, experimental testing, and numerical simulations, yielding a wealth of research results. However, the aforementioned studies lack a comprehensive investigation into the flexible characteristics of tail fins, and the majority of motion studies have focused on conventional symmetric motion patterns. To comprehensively explore the effects of tail fin structure and motion patterns on propulsion performance, this study adopts the dolphin as a bionic model, analyzing the structural characteristics of dolphin tail fins (including geometric and material properties). Through simulating the motion patterns of dolphins, a bidirectional fluid–structure interaction (FSI) numerical simulation is conducted to systematically investigate the hydrodynamic performance of bionic dolphin tail fins under different structural geometries, material properties, and motion patterns. The findings provide new insights for the design of bionic propulsion systems and offer practical guidance for their application in various motion scenarios.

## 2. Models and Methods

### 2.1. Geometric Model of the Tail Fin

This study references biological dolphin tail fin specimens (Figure 1a) to establish a bionic dolphin tail fin model. The 3D model of the tail fin is shown in Figure 1b, with a chord length C = 0.138 m, span E = 0.432 m, and a projected area S = 0.03 m^2^. According to the literature [31], the airfoil cross-section of the tail fin model is assumed to be NACA0021. To minimize the interference of the tail fin with the flow field during numerical simulations, the tail fin model was subjected to smoothing treatment.

### 2.2. Tail Fin Motion Model

In the process of steady-state dorsal–ventral propulsion, the trajectory of the dolphin’s tail fin can be considered as a sinusoidal curve (as shown in Figure 2), and this curve exhibits symmetry in both time and space (along the body’s longitudinal axis). It is generally believed that the movement of the tail fin is a combination of pitching motion and heaving motion. Heaving motion primarily generates the propulsive strokes, while the pitching motion of the tail fin around the tail base joint primarily provides an appropriate angle of attack for the propulsive strokes.

In numerical simulations, we convert the forward speed of the tail fin into incoming flow velocity [26], which can improve solution accuracy and efficiency. The motion model of the tail fin is shown in Figure 3, where the incoming flow velocity in the y+ direction is V, and the dorsal–ventral motion of the tail fin is decomposed into pitching motion around the *x*-axis and heaving motion along the *z*-axis. Let z(t) and θ(t) represent the position and orientation, respectively, of the tail fin at time t. Therefore, the periodic kinematics of the tail fin can be expressed using sine functions:(1)$$z(t)=z_{0}\sin(2\pi ft)$$(2)$$\theta (t)=\theta_{0}\sin(2\pi ft-\varphi )$$
where f represents the motion frequency of the tail fin, $z_{0}$ and $\theta_{0}$ denote the maximum amplitude and pitch angle, respectively, and φ represents the phase difference between heaving motion and pitching motion. According to related literature [31], a phase difference of 90° in dolphin tail fin motion yields the highest efficiency.

To quantitatively assess the propulsion performance of the bionic tail fin, non-dimensional thrust coefficient $C_{y}(t)$, non-dimensional lateral force coefficient $C_{z}(t)$, and moment coefficient $C_{m}(t)$ are defined based on the simulation model’s XYZ coordinate system as follows:(3)$$C_{y}(t)=\frac{2F_{y}(t)}{\rho_{f}SV^{2}}\;C_{z}(t)=\frac{2F_{z}(t)}{\rho_{f}SV^{2}}\;C_{m}(t)=\frac{2M_{x}(t)}{\rho_{f}SV^{2}}$$
where $F_{y}(t)$ represents thrust, $F_{z}(t)$ represents lateral force, $M_{x}(t)$ represents the instantaneous moment about the *x*-axis in pitching motion, and ρ is the density of the fluid medium. From the perspective of energy conversion, the input power $P_{in}$ is used to overcome the lateral force $F_{z}(t)$ and moment $M_{x}(t)$, while the output power $P_{out}$ can be defined as the energy from thrust $F_{y}(t)$ pushing the tail fin forward at velocity V.(4)$$P_{out}=\frac{1}{T}\int_{0}^{T}F_{y}(t)Vdt$$(5)$$P_{in}=\frac{1}{T}\int_{0}^{T}[F_{z}(t)z^{\prime }(t)+M_{x}(t)\theta^{\prime }(t)]dt$$
where $z^{\prime }(t)$ and $\theta^{\prime }(t)$ represent the velocities of the tail fin’s heaving motion along the *z*-axis and pitching motion around the *x*-axis, respectively.

The average thrust coefficient $C_{T}$, average pitching moment coefficient $C_{M}$, and the average input power coefficient $C_{P}$ are defined as follows:(6)$$C_{T}=\frac{1}{T}\int_{0}^{T}C_{y}(t)dt$$(7)$$C_{M}=\frac{1}{T}\int_{0}^{T}C_{m}(t)dt$$(8)$$C_{p}=\frac{P_{in}}{0.5\rho_{f}SV^{3}}$$

The propulsion efficiency η of tail fin motion can be expressed as the ratio of the average thrust coefficient $C_{T}$ to the average input power coefficient $C_{P}$.(9)$$\eta =\frac{P_{out}}{P_{in}}=\frac{C_{T}}{C_{P}}$$

### 2.3. Introduction to Numerical Methods

The Reynolds-averaged Navier–Stokes (RANS) model, continuity equation, and momentum equations are as follows [32]:(10)$$\frac{\partial \rho_{f}}{\partial t}+\nabla \cdot (\rho_{f}U)=0$$(11)$$\frac{\partial (\rho_{f}U)}{\partial t}+\nabla (\rho_{f}UU)=-\nabla \cdot P+\nabla \cdot \tau +\rho_{f}g$$
where in the effective shear stress tensor of the fluid τ:(12)$$\tau =(\mu (\nabla U+{\left(\nabla U\right)}^{T})-\frac{2}{3}(\nabla \;\cdot \;U)I)$$
where U represents fluid velocity, P represents fluid pressure, t represents time, g represents gravitational acceleration, and I represents the identity matrix.

The turbulence model selected is the SST model [33],(13)$$\begin{array}{l} \;\frac{\partial (\rho_{f}k)}{\partial t}+\nabla \;\cdot \;(\rho_{f}Uk)\\ =\nabla \;\cdot \;((\mu +\frac{\mu_{t}}{\sigma_{k}})\nabla k)+P_{k}-\rho_{f}\beta^{*}k\omega ,\\ \frac{\partial (\rho_{f}k)}{\partial t}+\nabla \;\cdot \;(\rho_{f}U\omega )=\nabla \;\cdot \;((\mu +\frac{\mu_{t}}{\sigma_{k}})\nabla \omega )\\ +P_{\omega }-\rho_{f}\beta \omega^{2}+2.336(1-F_{1})\frac{\rho_{f}}{\omega }\frac{\partial k}{\partial x_{j}}\frac{\partial \omega }{\partial x_{j}} \end{array}$$

Turbulent viscosity coefficient $\mu_{t}$:(14)$$\mu_{t}=\frac{a_{1}\rho_{f}k}{\max(a_{1}\omega ,SF_{2})}$$
where k represents turbulent kinetic energy, $P_{k}$ represents the production term of turbulent kinetic energy caused by the mean velocity gradient, represents the dissipation rate of turbulent kinetic energy, $P_{\omega }$ is generated by the special turbulent kinetic energy dissipation rate ω, S represents the tensor modulus of the mean strain rate, $F_{1},F_{2}$ represents the mixing function, and β and $\beta^{*}$ represent coefficients in the turbulence model, while $a_{1}$ represents the turbulence model constants.

The following equations are involved in the solid domain solution. Solid transient dynamic equilibrium equations [34]:(15)$$\rho_{s}\frac{d^{2}d_{s}}{dt^{2}}=\nabla \cdot \sigma_{s}+\rho_{s}g$$

Solid strain compatibility equations:(16)$$\nabla \times \Gamma_{s}\times \nabla =0$$
where $d_{s}$ represents the displacement vector, $\sigma_{s}$ represents the Cauchy stress tensor, g represents the gravity acceleration vector, and $\rho_{s}$ represents the solid density.

According to the generalized Hooke’s law, the strain tensor $\Gamma_{s}$ is:(17)$$\Gamma_{s}=\frac{1+v_{s}}{E}\sigma_{s}-\frac{v_{s}}{E}ITr(\sigma_{s})$$
where E represents Young’s modulus, $\nu_{s}$ represents Poisson’s ratio, and $Tr(\sigma_{s})$ represents the trace of the stress tensor.

This study employs the Ansys bidirectional implicit interaction method for bidirectional fluid–structure interaction simulation (Figure 4). Fluent and Transient Structural perform implicit iterative solutions for the flow field and structural field, respectively, at each time step, and data are transferred through a pre-defined fluid–structure interaction interface. Waiting for the data transmitted at the fluid–structure interaction interface to converge before proceeding to the next time step until the final computation is completed, this method greatly enhances solution speed and convergence while ensuring solution accuracy.

The above process requires two mathematical boundary conditions based on stress continuity and velocity continuity. Stress continuity boundary conditions for solid mechanics calculations [34]:(18)$$-pn+\tau \;\cdot \;n=\sigma_{s}\;\cdot \;n$$

The left side of the equation consists of the effective components of pressure and shear transmitted from the fluid domain on the FSI surface, while the right side consists of the effective components of the solid Cauchy stress tensor on the FSI surface.

Velocity continuity boundary conditions for fluid calculations:(19)$$u=\frac{dd_{s}}{dt}$$

The left side of the equation represents the fluid velocity on the FSI surface, while the right side represents the displacement velocity of the solid FSI surface. In addition to solving the fluid flow equations, this velocity is also used to compute and update the moving mesh.

### 2.4. Computational Domain and Mesh Generation

The numerical simulation involves the rigid and flexible deformation motions of the surface mesh of the study object, which lead to changes in the topological structure of the computational domain mesh. Therefore, a dynamic mesh method is required to update the internal computational mesh in real time. The spring smoothing method is employed to control the overall deformation of the computational domain mesh associated with the deformed regions. For meshes with significantly degraded quality due to excessive local deformation, a local mesh reconstruction method is utilized. The dolphin’s dorsal–ventral motion (including heaving and pitching motions) is simulated by applying kinematic joints to the tail fin model within the solid module.

As shown in Figure 5a, the computational domain in this study is a rectangular domain of 14C × 4C × 8C, the SST k − ω model is chosen for the turbulence model, with the left boundary of the rectangle serving as the fluid inlet where the inflow velocity is fixed, and the right boundary as the pressure outlet with pressure set to 0. The origin of the coordinate system is located at the center of the leading edge of the tail fin, with the distance between the leading and trailing edges of the tail fin and the front and rear boundaries in a ratio of 2:11, allowing sufficient space for the wake vortex structure to be displayed. To avoid interference from ground effects, the center of the tail fin’s leading edge is situated at a distance of 2C from the upper and lower boundaries. The tail fin surface is set to a no-slip condition, while the surrounding walls are set to slip conditions. The mesh of the fluid domain is shown in Figure 5b. In fluid calculations, the mesh near the tail fin surface needs to be refined, with sparse mesh sizes of 20 mm and fine mesh sizes of 2 mm. A tetrahedral sparse mesh with a size of 5 mm is used for partitioning the solid regions, which helps to improve computational efficiency. Given that the tail fin motion involves flexible deformations, a fine tetrahedral mesh is used. A grid-independent algorithm is adopted, limiting the minimum mesh size to 0.5 mm, with additional mesh refinement applied to the edges of the tail fin.

## 3. Verification of Numerical Methods

### 3.1. Sensitivity Test

Before conducting numerical simulations, it is essential to validate grid independence and the sensitivity of the time step. An appropriate number of grid cells and a suitable time step can make the computational results more accurate and reliable. The parameters for this case are as follows: f=0.5 hz, $z_{0}=0.5\;C$, θ=25°, φ=90°, V=0.2 m/s.

In theory, the smaller the time step, the higher the solution accuracy; however, smaller time steps result in larger computational loads and lower solution performance. In this section, three time steps (0.001, 0.005, and 0.01) were selected for numerical simulations. Figure 6 presents the variation curves of instantaneous thrust coefficients and instantaneous lateral force coefficients under the three time steps. The simulation results for 0.001 and 0.005 are nearly identical, while the simulation result for 0.01 shows some discrepancies compared to the results for 0.001 and 0.005. Therefore, in the subsequent simulations, the time step was set to 0.005.

### 3.2. Independence Verification

Further research was conducted on the number of fluid domain grids by using three different computational meshes for grid independence verification, namely coarse mesh, medium mesh, and fine mesh. Figure 7 presents the variation curves of instantaneous thrust coefficients and instantaneous lateral force coefficients under the three mesh densities. It can be seen that the curves for the medium mesh and fine mesh are almost identical, while the coarse mesh results exhibit some discrepancies compared to the first two. Considering computational accuracy and performance, the medium mesh with a grid count of 3,199,585 was selected for the subsequent simulations.

### 3.3. Reliability Verification

To verify the reliability of the simulation methods used in this study, a series of hydrodynamic experiments simulating dolphin tail fin motion were conducted. The hydrodynamic experiment platform, shown in Figure 8, consists of a water tank, a SCARA robot, an SRI multi-axis force sensor, and a bionic dolphin tail fin. The water tank provides a static water environment, and the SCARA robot is programmed to drive the bionic tail fin in sinusoidal motion. The sensor model used is the M3816D, developed by SRI International. During force measurements, the error typically does not exceed ±0.1 N, with a resolution of 0.01 N. For torque measurements, the error is within ±0.01 Nm, and the resolution reaches 0.001 Nm. This SRI multi-axis force sensor is used to measure the thrust along the *y*-axis, the lateral force along the *x*-axis, and the moment around the *z*-axis. A data acquisition card (M8128) is used to measure and process force sensor data at a rate of 1000 samples per second. The bionic tail fin was fabricated using 3D printing technology with black PLA resin as the material. The material properties include a Young’s modulus of 3.0 GPa and a Poisson’s ratio of 0.35, and no deformation occurred during the experiment, indicating that the tail fin has sufficient stiffness. The heaving motion ranges from $z_{0}=0.1-1.0\;C$, and the other parameters are as follows: f=0.5 Hz, θ=25°, φ=90°, V=0.2 m/s.

Figure 9a,b shows the time-varying curves of $C_{x}$ and $C_{y}$ during the motion cycle at $z_{0}=0.5\;C$ for both simulation and experiment. It can be observed that at $z_{0}=0.5\;C$, while there is a certain degree of error between the simulation and experimental curves, the overall variation trends are consistent. From Figure 10, it can be seen that at $z_{0}=0.5\;C$, the average thrust coefficient error between the simulation and experiment is approximately 30%. However, as the amplitude of the undulating motion increases, the error between the two decreases, reaching only 4.15% at $z_{0}=\;1\;C$. This indicates that the numerical method employed in this study provides the required level of accuracy for the solution.
(1)In the experiment, the connection between the tail fin and the robot interacts with the water, affecting the measurement of mechanical parameters, especially when the amplitude is small. As shown in Figure 10, the discrepancies between experimental and simulation values are more significant when $z_{0}/C\;\leq \;0.4$.(2)In the simulation, periodic boundary conditions were applied, whereas in the experiment, the finite boundaries of the water tank may have caused reflection and interference effects in the flow field near the tail fin.(3)The PLA material model used in the simulation is based on the typical mechanical parameters reported in the literature. However, the actual PLA material used in the experiment may exhibit non-uniformity (e.g., insufficient interlayer bonding strength in 3D-printed components), leading to discrepancies between the mechanical response of the material in the experimental results and the simulation model.(4)During the measurement of thrust and flow fields in the experiment, the resolution and installation accuracy of the equipment may introduce uncertainties. To minimize the impact of such errors, multiple experimental measurements were conducted to reduce random errors.

## 4. Results and Discussions

Vortices play a crucial role in the propulsion process of the tail fin. The vortices generated during the oscillation of the tail fin are not merely disturbances in the fluid; they directly influence the generation of thrust, efficiency, and overall propulsion effectiveness of the tail fin. When the tail fin oscillates, a velocity difference occurs in the fluid on both sides of the tail fin, leading to the formation of vortices. These vortices, in turn, exert reactive forces on the fluid, pushing the tail fin forward and thereby generating thrust. The formation and shedding of vortices actually constitute one of the primary mechanisms for thrust generation during the oscillation of the tail fin.

As shown in Figure 11, we analyze the vortex structures formed during the oscillation process of bionic dolphin tail fins within one motion cycle. At (N + 1/4)T, the vortex structure behind the tail fin is relatively weak, and the tail fin is at the upper extreme position in the oscillation direction. At this time, the generation of tail fin vortices is minimal, new vortices have not yet formed, and previously formed vortices gradually shed and enter the wake. The rate of vorticity generation is slow, and after vortex shedding, the fluid gradually stabilizes, resulting in minimal force from the tail fin on the fluid. At (N + 1/2)T, the tail fin reaches the middle position, and a strong vortex pair is generated behind the tail fin. At this time, the rate of vorticity generation is the fastest, the tail fin’s movement speed reaches its maximum, and significant shear effects are produced. The strong vortex pair structure is clear and begins to gradually shed, with the most significant fluid disturbances and the most evident tail fin water-pushing effect. The intense vortex field supports the peak thrust. At (N + 3/4)T, the tail fin moves to the extreme position on the other side, and the vortex pair structure behind the tail fin gradually moves away from the tail fin, entering the wake area. The generation of new vortices weakens, resulting in a sparser vortex field. Thrust approaches zero or becomes negative again, and the system enters the reverse switching phase. At (N + 1)T, after the tail fin switches direction and accelerates, vortex generation gradually increases, and the vortex field becomes clearer and stronger. Vortices gradually form, thrust increases, the tail fin resumes pushing the water flow, and new vortex pairs are generated. At this moment, the tail fin returns to the middle position again.

### 4.1. Material Properties of Dolphin Tail Fin

According to the study by Sun et al. [35], the structure of the dolphin tail fin is similar to a sandwich composite beam, primarily composed of an external ligament layer (LL) with a high tensile modulus and an internal dense connective tissue (DCT) with a high compressive modulus, as shown in Figure 12 [36]. The study found a linear relationship between the thickness of the external ligament layer $T_{LL}$ and the total thickness of the tail fin $T_{F}$ [37].(20)$$T_{LL}=0.843+0.11T_{F}$$(21)$$E_{eq}=\frac{2\left\{\int_{0}^{T_{F}/(2-T_{LL})}E_{DCT}x^{2}dx+\int_{T_{F}/(2-T_{LL})}^{T_{F}/2}E_{LL}x^{2}dx\right\}}{\int_{-T_{F}/2}^{T_{F}/2}x^{2}dx}$$

Sun et al. [35] conducted tensile and compression experiments by collecting biological samples of deceased dolphin tail fin, obtaining the Young’s moduli $E_{LL}=166.54\;\mathrm{Mpa}$ and $E_{DCT}=12.05\;\mathrm{Mpa}$ in the spanwise direction for LL and DCT, respectively. Using Equation (21), we can obtain the equivalent Young’s modulus $E_{eq}=108\;\mathrm{Mpa}$ of the biological dolphin tail fin.

### 4.2. Effect of Structural Parameters on Propulsion Performance

To study the effect of structural parameters on the propulsion performance of bionic dolphin tail fins, in addition to the bionic tail fin with a variable-thickness structural shape mentioned above, we also designed an equal-thickness flat structural bionic dolphin tail fin. In terms of materials, besides the equivalent characteristic materials of the dolphin tail fin, we also selected PE materials and structurally rigid materials, with material properties shown in Table 1. Finally, simulation analyses were conducted on six types of tail fins: an equal-thickness structural steel tail fin (ES), equal-thickness PE tail fin (EM), equal-thickness bionic material tail fin (EF), variable-thickness structural steel tail fin (VS), variable-thickness PE tail fin (VM), and variable-thickness bionic material tail fin (VF). To eliminate the influence of other physical factors, the following parameters were set in the numerical simulation: f=0.5 Hz/0.8 Hz/1 Hz, $z_{0}=C$, θ=25°, φ=90°, V=0.7 m/s.

As shown in Figure 13 and Table 2, we analyze the thrust coefficients and propulsion efficiencies of six types of bionic dolphin tail fins at different motion frequencies. At a motion frequency of 0.5 Hz, material properties have little impact on the propulsion performance of tail fins with identical structural shapes. At motion frequencies of 0.8 Hz and 1 Hz, the propulsion performance rankings of the tail fins are VF > VM > VS and EF > EM > ES, respectively. At these motion frequencies, regardless of equal thickness or variable thickness, bionic-material tail fins exhibit better propulsion performance. This conclusion, from a bionic perspective, aligns with the oscillation characteristics of dolphins. At motion frequencies of 0.8 Hz and 1 Hz, the Strouhal numbers of bionic dolphin tail fins are 0.32 and 0.4, respectively, which fall within the real dolphin’s Strouhal number range (0.2–0.45). Under the same motion frequency, variable-thickness tail fins with identical material properties have higher average thrust coefficients than equal-thickness tail fins. The analysis shows that this is due to the leading-edge suction effect: when fluid flows over the leading edge of a variable-thickness tail fin, the flow velocity in that region increases, thereby creating a low-pressure zone, generating leading-edge suction, and providing part of the thrust to propel the tail fin forward. In the red-circled areas of Figure 13a–c, the tail fin is in a horizontal position relative to the motion direction, and the effect of leading-edge suction on instantaneous thrust is most apparent. In summary, we believe that the structural parameters of VF are more conducive to enhancing the propulsion performance of bionic dolphin tail fins.

### 4.3. Effect of Asymmetric Motion Modes on Propulsion Performance

Based on the above research findings, we take VF as the subject to study the effect of asymmetry in dolphin motion frequency and amplitude on propulsion performance. We define the frequency ratio F and amplitude ratio H, with values as shown in Table 3. Among them, all frequency ratios F correspond to equal tail fin oscillation periods and amplitudes; all amplitude ratios H correspond to equal tail fin oscillation periods and peak-to-peak amplitudes. In the numerical simulations, the parameters for symmetric motion are set consistently with the above, with f=1 Hz.

#### 4.3.1. Analysis of Frequency Ratio Effects

Figure 14 shows the influence curves of frequency ratio F on hydrodynamic parameters. From Figure 14b, within the examined range, the average thrust coefficient $C_{T}$ exhibits a roughly symmetrical distribution with respect to the frequency ratio F, and the larger the frequency difference between the upward swing and downward swing of the tail fin, the greater the $C_{T}$. From Figure 14d, when performing symmetric tail oscillation (F=5/5), the average pitching moment coefficient $C_{M}$ is minimized, indicating that symmetric motion suppresses pitching moments, and the closer F is to 1, the better the suppression effect. From Figure 14a,c, the peak moments of the instantaneous thrust coefficient and instantaneous pitching moment coefficient occur during the high-frequency tail oscillation phase. Asymmetric motion exacerbates the amplitude of the instantaneous thrust coefficient peaks, which is the main factor for thrust enhancement. However, this also increases pitching moments, affecting the stability of motion.

#### 4.3.2. Analysis of Amplitude Ratio Effects

Figure 15 shows the influence curves of amplitude ratio H on hydrodynamic parameters. From Figure 15b, it can be seen that the average thrust coefficient $C_{T}$ also approximately exhibits a symmetrical distribution with respect to the amplitude ratio H, and asymmetric amplitude motions help to increase thrust. $C_{T}$ increases as the difference in amplitude between the upward and downward swings increases, and when H<1, it is more favorable for enhancing thrust, due to the influence of the water depth pressure gradient. The trend of $C_{M}$ changes is similar to that of $C_{T}$. From Figure 15a,c, it can be seen that the abrupt changes in the instantaneous thrust coefficient curves and instantaneous pitching moment coefficient curves occur during the large-amplitude phase; the greater the difference in amplitude between the upward and downward swings, the more drastic the curve changes. These drastic changes enhance thrust but also exacerbate motion instability.

Based on the above research findings, symmetric motion is suitable for dolphins in steady horizontal cruising, while asymmetric motion is more suitable for movement in the vertical plane. The average pitching moment of asymmetric amplitude motion is greater than that of asymmetric frequency motion, indicating that asymmetric amplitude motion is more suitable for dolphins’ movement in the vertical direction. This conclusion also aligns with the actual movement behavior of dolphins.

Therefore, to avoid the impact of asymmetric motion on the stability of the robot during cruising, we suggest introducing a closed-loop feedback control system for underwater bionic robots. Sensors can be used to monitor the dynamic behavior of the tail fin in real time, and motion parameters can be adjusted using PID control or adaptive control algorithms to prevent asymmetric motion during cruising. Furthermore, a multi-mode switching control algorithm should be developed to allow switching between motion modes based on operational requirements. This enables optimized motion in different states and effectively prevents the dynamic instability caused by asymmetric motion during cruising.

## 5. Conclusions

In this study, we used a bidirectional fluid–structure interaction (FSI) technique to numerically analyze the hydrodynamic characteristics of a biomimetic dolphin tail fin simulating the dorsal–ventral motion of the dolphin tail fin. The results show that altering the flexibility of the tail fin can improve its propulsion performance, particularly with tail fins made from biomimetic materials, which exhibit outstanding performance within the Strouhal number (St) range of biological dolphins. At the same motion frequency, the propulsion of thickened tail fins with identical material properties generally outperforms that of uniform-thickness tail fins. This suggests that biomimetic structures have more advantages when it comes to designing underwater propulsion devices. Asymmetric motion modes have a boosting effect but also increase the instability of the movement, making them only suitable for dolphins’ motion in the vertical plane.

This study not only reveals the core factors behind the propulsion mechanism of the tail fin but also provides important guidance for the optimization of biomimetic dolphin tail fins, with broad application potential, especially in the design of underwater robots and the development of biomimetic propulsion systems. Future research can further investigate the impact of different material and structural combinations on propulsion performance, along with experimental validation in real-world scenarios, to achieve more efficient and stable biomimetic propulsion devices. Additionally, integrating intelligent control technologies with biomimetic tail fins is another promising research direction to explore in the future.

## Figures and Tables

**Figure 1 biomimetics-10-00059-f001:**
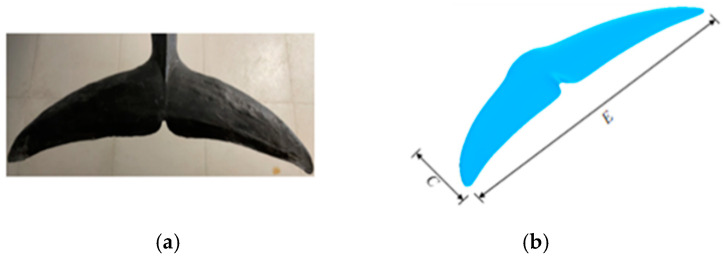
(**a**) Dolphin tail fin specimen and (**b**) bionic dolphin tail fin model.

**Figure 2 biomimetics-10-00059-f002:**
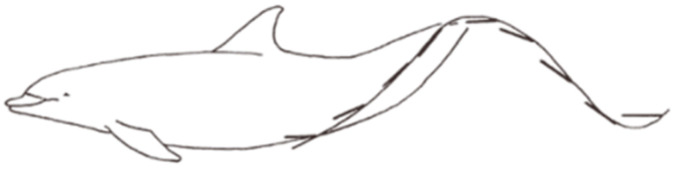
Schematic diagram of the steady-state dorsal–ventral motion trajectory of the dolphin.

**Figure 3 biomimetics-10-00059-f003:**
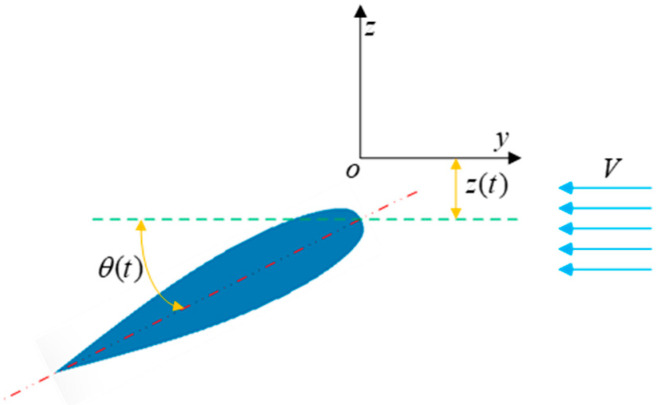
Tail fin motion model.

**Figure 4 biomimetics-10-00059-f004:**
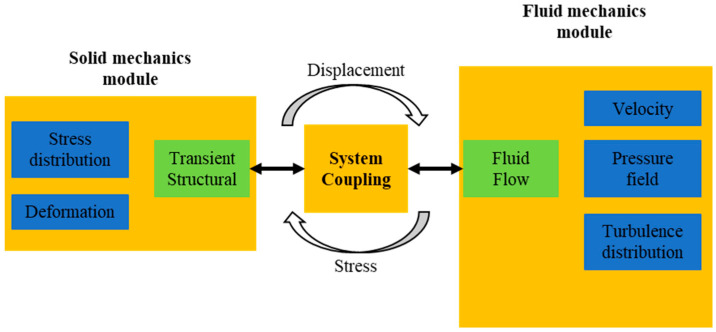
Data flow diagram of Ansys bidirectional implicit fluid–structure interaction simulation analysis.

**Figure 5 biomimetics-10-00059-f005:**
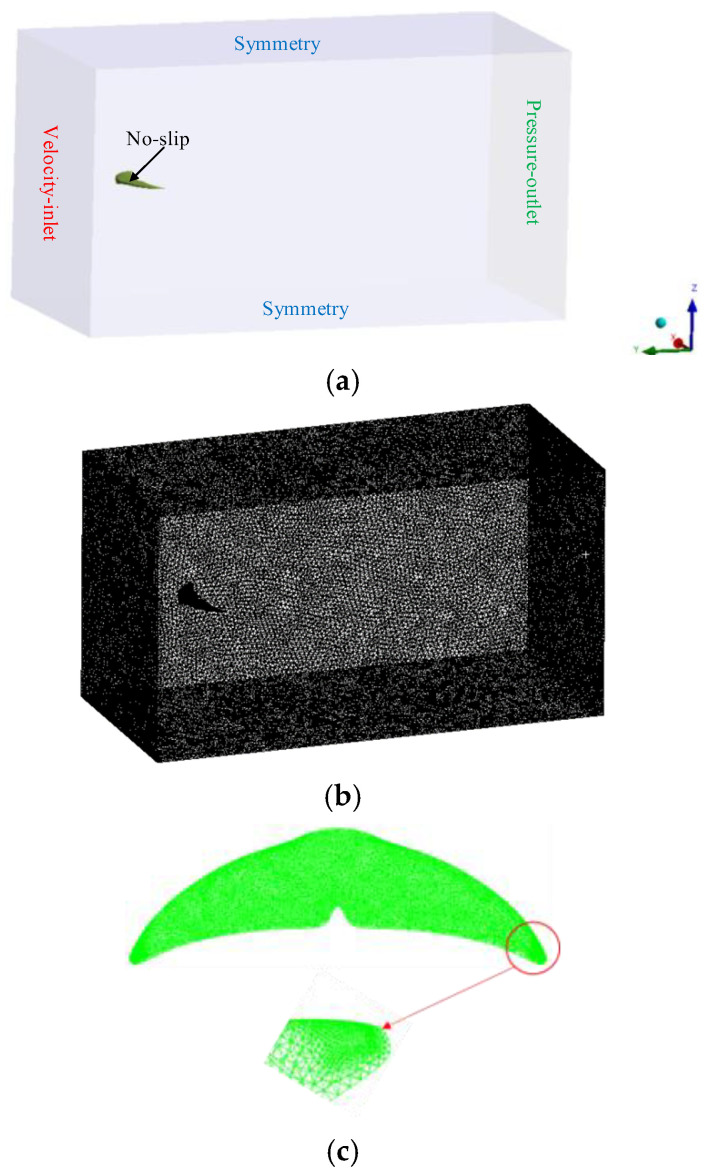
(**a**) Computational domain model, (**b**) fluid domain mesh generation, and (**c**) solid domain mesh generation.

**Figure 6 biomimetics-10-00059-f006:**
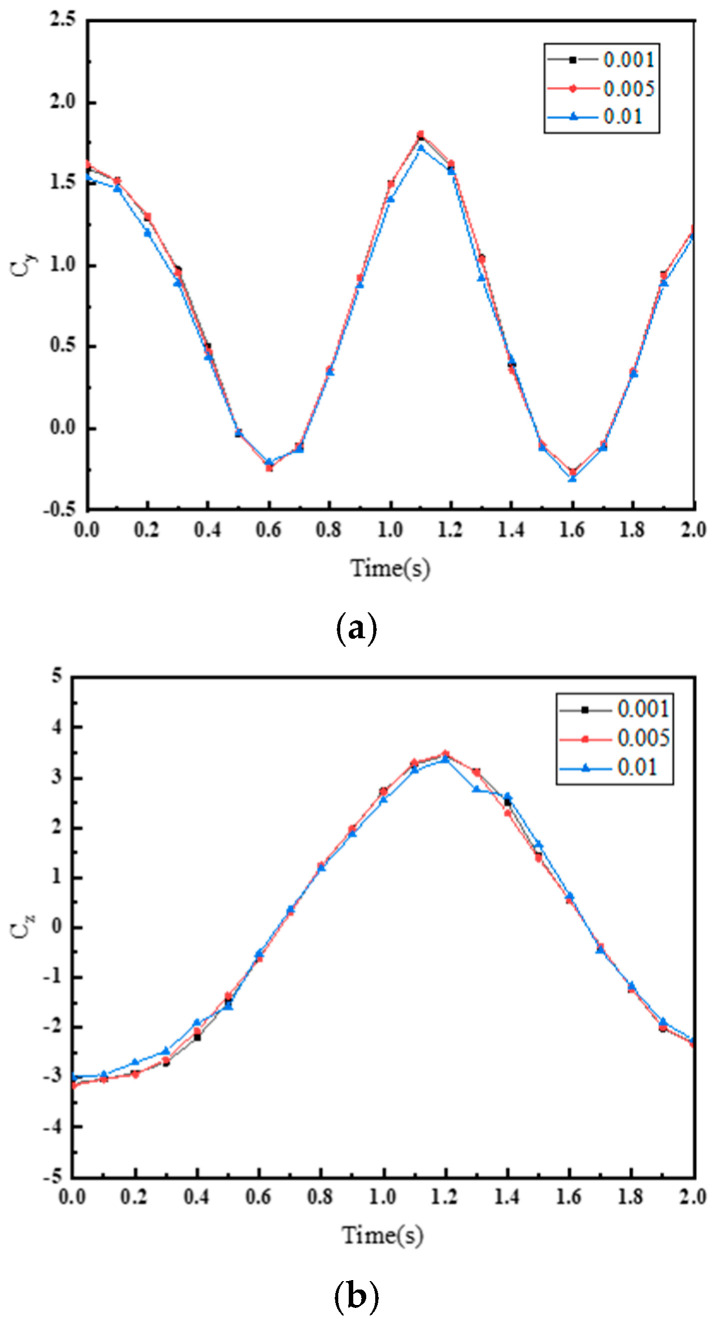
Variation of (**a**) instantaneous thrust coefficient and (**b**) instantaneous lateral force coefficient with time for different time steps.

**Figure 7 biomimetics-10-00059-f007:**
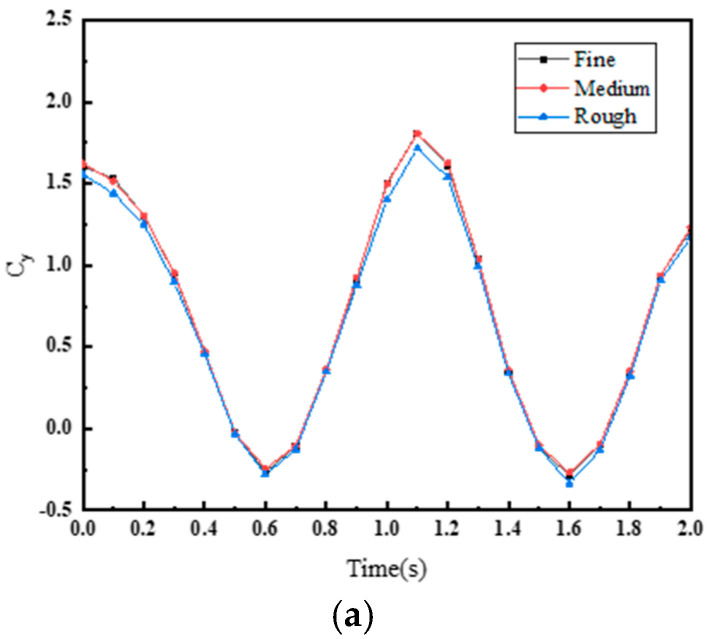
Variation of (**a**) instantaneous thrust coefficient and (**b**) instantaneous lateral force coefficient with time for different mesh sizes.

**Figure 8 biomimetics-10-00059-f008:**
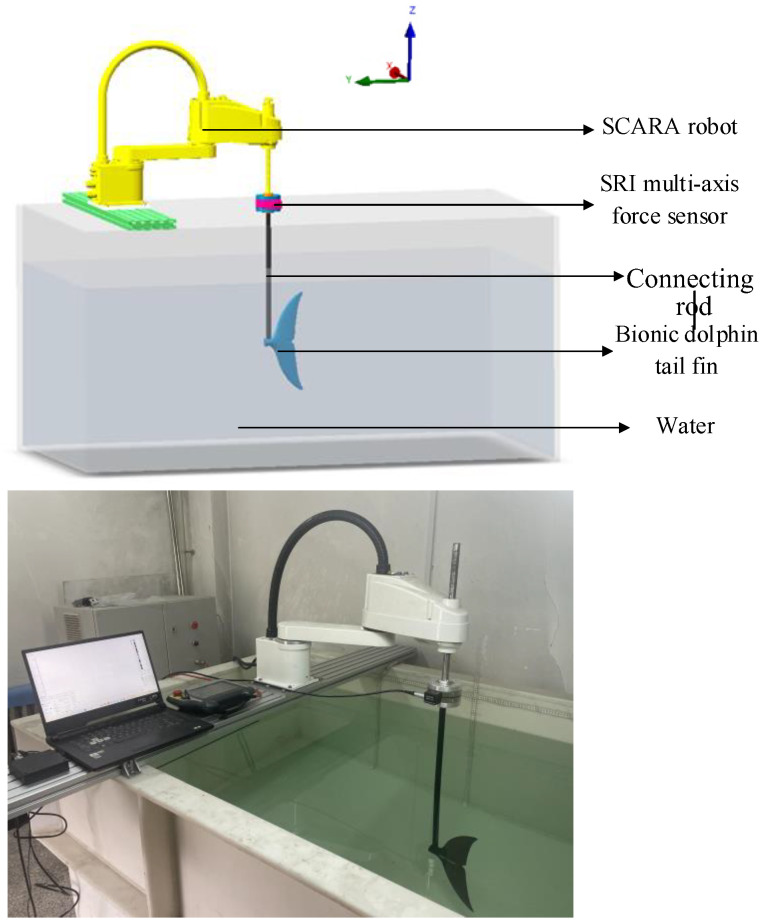
Experimental platform.

**Figure 9 biomimetics-10-00059-f009:**
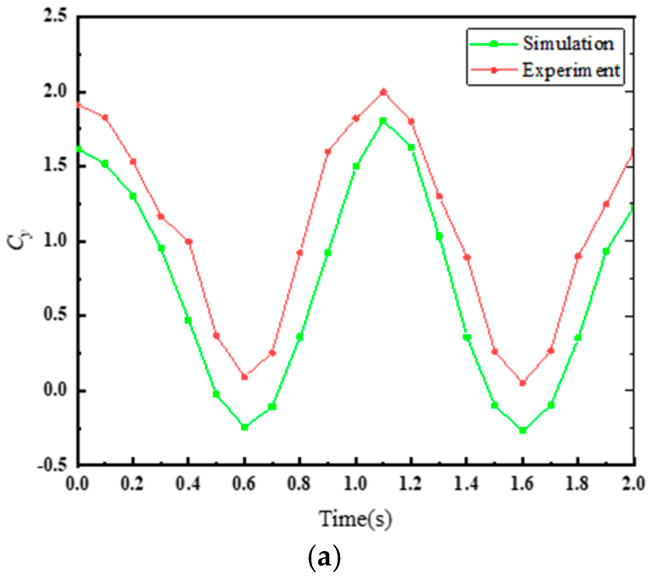
Time-varying curves of (**a**) $C_{y}$ and (**b**) $C_{z}$ during the motion cycle at $z_{0}=0.5\;C$ for simulation and experiment.

**Figure 10 biomimetics-10-00059-f010:**
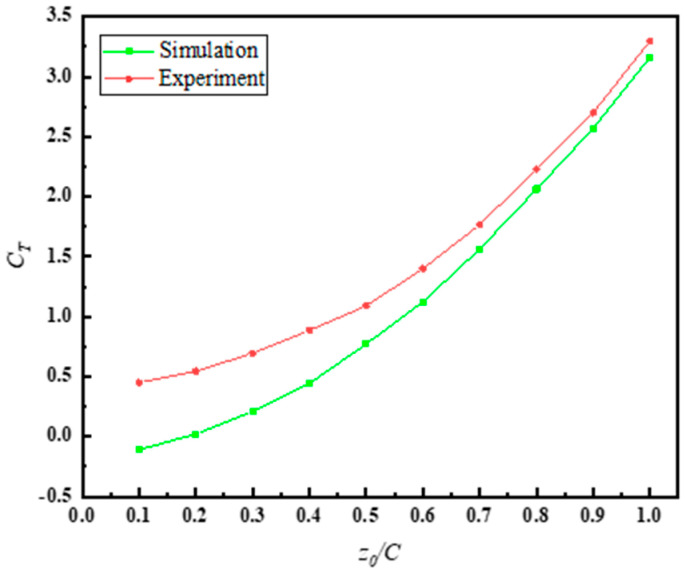
Variation curve of the average thrust coefficient $C_{T}$ with $z_{0}/C$ for simulation and experiment.

**Figure 11 biomimetics-10-00059-f011:**
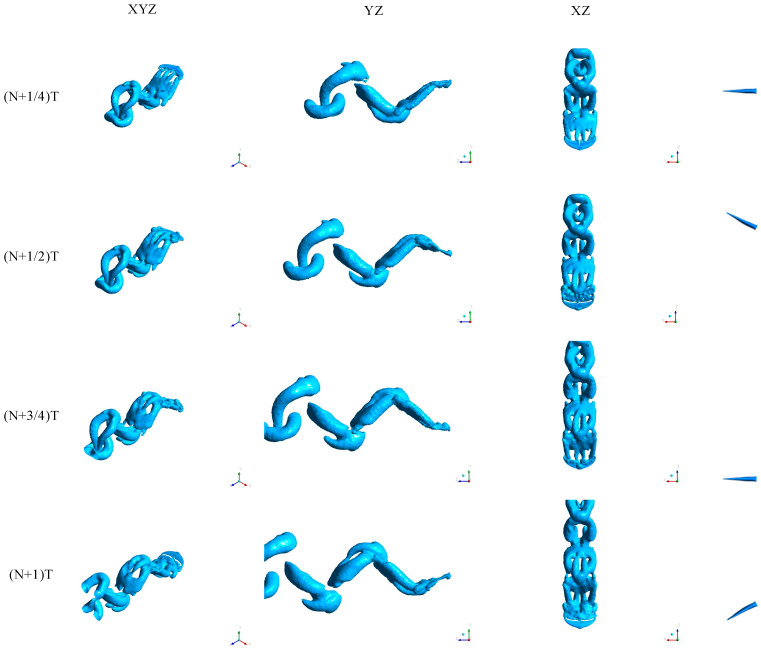
Vortex structure.

**Figure 12 biomimetics-10-00059-f012:**
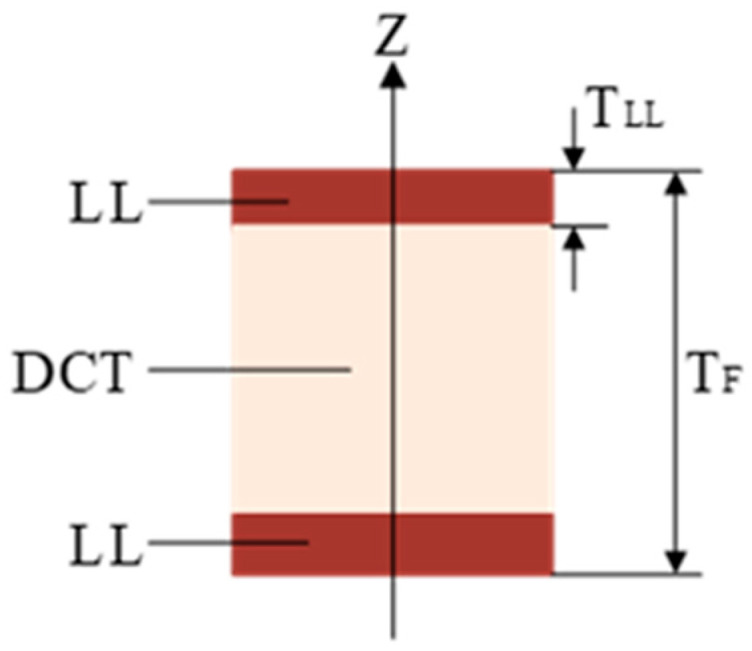
Schematic diagram of the cross-section of the dolphin tail fin.

**Figure 13 biomimetics-10-00059-f013:**
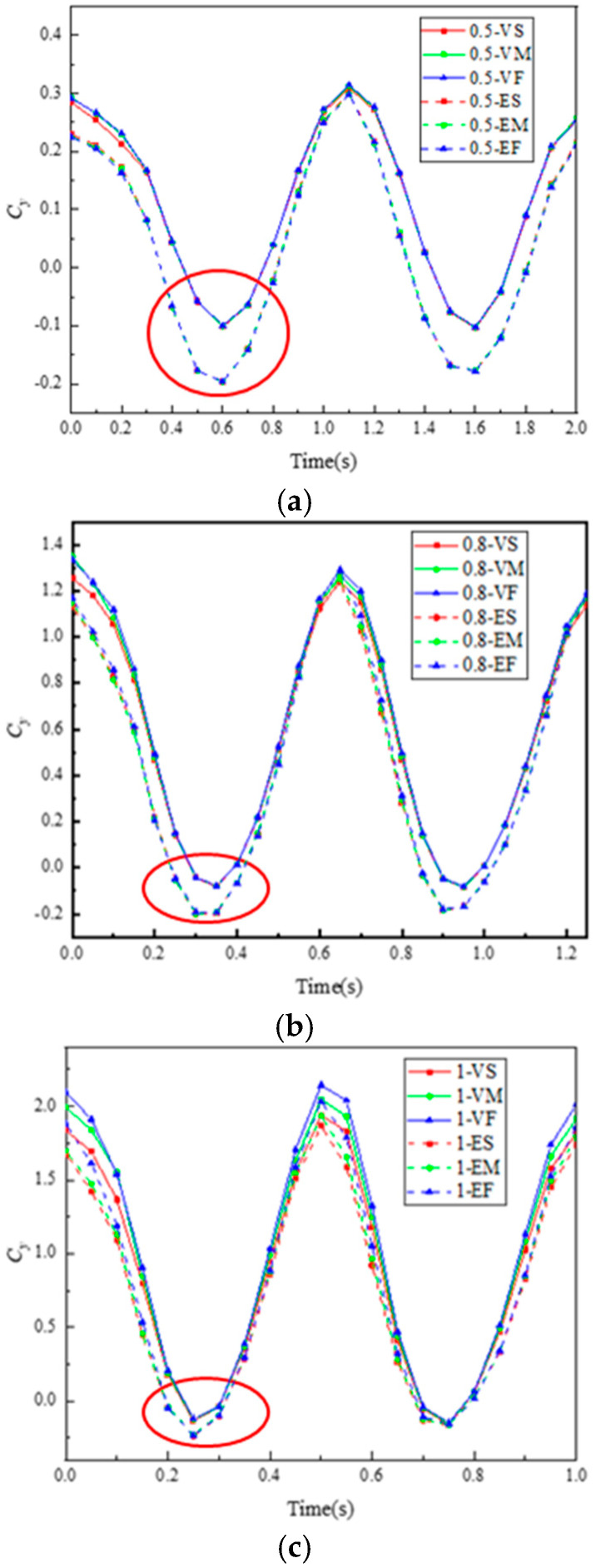
Variation curves of thrust coefficients of each tail fin at different motion frequencies: (**a**) 0.5 Hz, (**b**) 0.8 Hz, (**c**) 1 Hz; and (**d**) propulsion efficiency.

**Figure 14 biomimetics-10-00059-f014:**
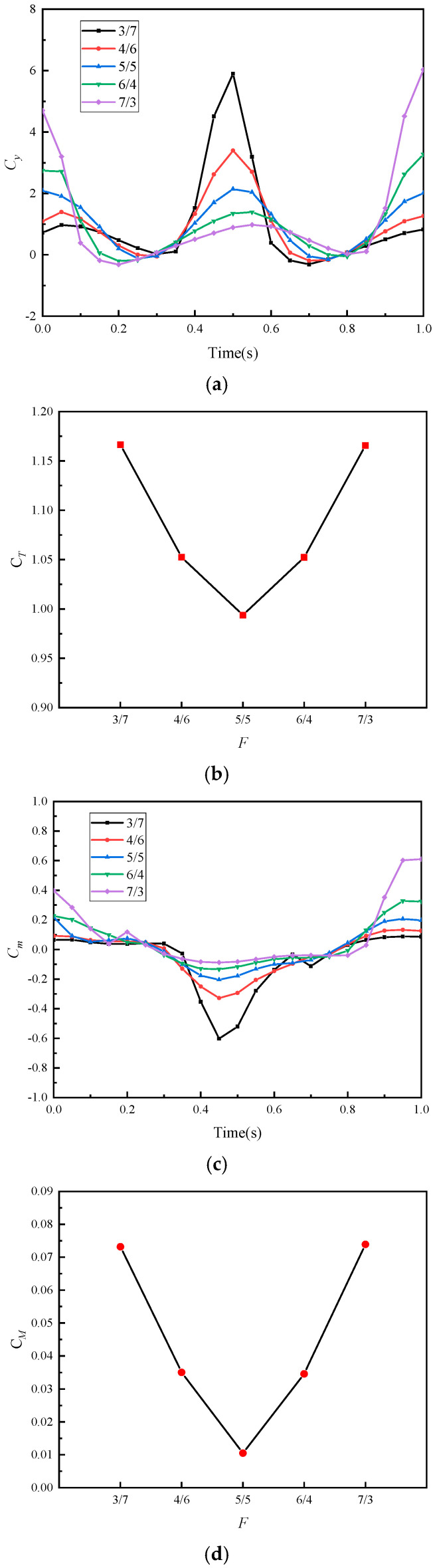
Hydrodynamic parameter figures under different frequency ratios F: (**a**) thrust coefficient, (**b**) average thrust coefficient, (**c**) pitching moment coefficient, and (**d**) average pitching moment coefficient.

**Figure 15 biomimetics-10-00059-f015:**
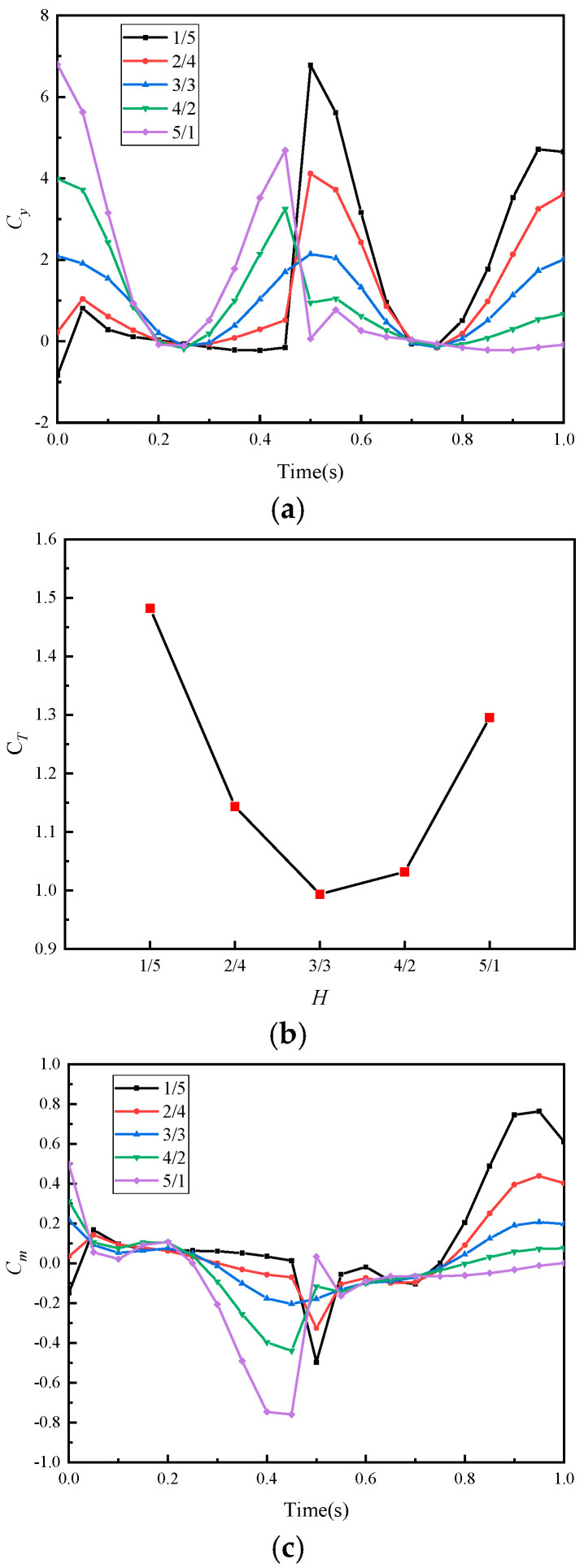
Hydrodynamic parameter figures under different amplitude ratios H: (**a**) thrust coefficient, (**b**) average thrust coefficient, (**c**) pitching moment coefficient, and (**d**) average pitching moment coefficient.

**Table 1 biomimetics-10-00059-t001:** Material properties of the bionic tail fin.

	Young’s Modulus (Pa)	Poisson’s Ratio
Bionic material	1.08 × 10^8^	0.45
PE	5 × 10^8^	0.42
Structural steel	2 × 10^11^	0.3

**Table 2 biomimetics-10-00059-t002:** Average thrust coefficients of each tail fin at different motion frequencies.

Motion Frequencies (Hz)	Average Inference Coefficient C*_T_*					
	VF	VM	VS	EF	EM	ES
0.5	0.11178	0.11264	0.11331	0.04057	0.03989	0.03997
0.8	0.56607	0.5829	0.59252	0.45624	0.46011	0.46976
1	0.88663	0.94834	0.9937	0.74296	0.76869	0.81563

**Table 3 biomimetics-10-00059-t003:** Ratio parameter settings.

F	H
3/7, 4/6, 5/5, 6/4, 3/7	1/5, 2/4, 3/3, 4/2, 5/1

## Data Availability

All data generated or analyzed during this study are included in this published article.

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
