# Peer review of "Hydrodynamic Characteristics Study of Bionic Dolphin Tail Fin Based on Bidirectional Fluid–Structure Interaction Simulation"

_biomimetics, 2025, doi:10.3390/biomimetics10010059_

Round 1
Reviewer 1 Report
Comments and Suggestions for Authors
This study employs bidirectional fluid-structure interaction technology to investigate the hydrodynamic characteristics of bionic dolphin tail fins through simulations and experiments. The thrust coefficients and efficiency of tail fins made of equal thickness and variable thickness materials, including steel, PE, and bionic materials, were analyzed and compared under different motion frequencies. Results indicate that variable thickness bionic materials exhibit superior propulsion performance. This highlights the advantages of bionic structures in the design of underwater propulsion devices. Additionally, the study examines symmetric and asymmetric motion modes of the tail fins, finding that symmetric motion is suitable for stable cruising, while asymmetric motion is advantageous for vertical movements. These findings provide insights into the propulsion mechanisms of dolphin tail fins. The paper can be accepted if the following comments are addressed:
1. In Section 2.4, what is the number of grid divisions used in the study? Given the high boundary layer requirements of the SST model, how was the boundary layer around the tail fin discretized? Was a grid convergence analysis conducted?
2. In Section 3, the errors between the simulation and experimental results should be analyzed and quantified. While Figures 7 (a) and 8 demonstrate consistent trends in various coefficients, could the discretization of the boundary layer grid during the simulation potentially influence the results?
3. In Figure 6, the Cartesian coordinate system and the corresponding motion directions should be indicated. Why do the tail fin postures in the experiments and simulations differ?
4. In Figure 9, the image should be enlarged to clarify the position and posture of the tail fin, as they are not clearly discernible.
5. Histograms used in Figure 11 (b), (d), and (f), Figure 12 (b) and (d), Figure 13 (b) and (d). It is recommended to use a line chart with different line styles to represent the data or to present the information in a table for clarity.
6. The ticks and legends (e.g. 37, 46, 55) in Figures 12 and 13 must be corrected.
7. In Line 151, γ-axis is supposed to be x-axis.
Reviewer 2 Report
Comments and Suggestions for Authors
The focus on the effects of tail fin flexibility and structure is commendable. However, prior studies, such as “A Hydrofoil Propeller That Can Change Shape Like a Fish” and “Numerical and Experimental Investigation of Bio-Inspired Robot Propulsion and Manoeuvring,” have already addressed the functionality of fin flexibility. This puts the originality of the present study into question. The authors must at least reference these prior works and clarify what distinguishes their study to assert its uniqueness.
The lack of a thorough literature review undermines the justification for the study’s novelty. Consequently, the positioning of this research is weakened, and the ability to compare it with other studies is diminished.
Limitations in Experimental Validation
- While water tank experiments were conducted to validate the simulations, the experimental setup and scope may be insufficient. The differences between the experimental and simulation conditions, highlighted as causes of errors, are not discussed in sufficient detail.
Insufficient Discussion on Stability of Asymmetric Motion Modes
- The study shows that asymmetric motion modes enhance thrust but reduce dynamic stability. However, no strategies or control methods to mitigate this instability are proposed. This necessitates additional discussion.
Issues with Numerical Model Accuracy and Experimental Reproducibility
- The numerical model used to analyze fluid-structure interaction does not sufficiently address its limitations, such as turbulence modeling or the impact of mesh generation. This raises potential doubts about the accuracy and validity of the results.
- The study lacks detailed information to ensure the reproducibility of the experiments. Specifics on material properties, such as flexibility and rigidity, as well as measurement accuracy, are inadequately discussed.
Recommendations for the Authors
The authors must address these points explicitly in the manuscript. Without adequate explanations and discussions on these issues, the current version of the manuscript cannot be accepted for publication.
Comments on the Quality of English LanguageThere are some minor improvements (eliminating redundancies, adding logical connectives, etc.), but there are few areas where reviewers or readers will have difficulty understanding.
